# Observations of genetic differentiation between the fall armyworm host strains

**Rodney N. Nagoshi** [ID] *

Center for Medical, Agricultural and Veterinary Entomology, United States Department of Agriculture-Agricultural Research Service, Gainesville, Florida, United States of America

* rodney.nagoshi@usda.gov

## Abstract

The threat of invasive species is increasing with the expansion of global trade and habitat disruption. A recent example is the establishment of fall armyworm (FAW), a noctuid moth native to the Americas, into most of the Eastern Hemisphere with projections of significant economic losses on a global scale. The species has traditionally been subdivided into two populations that differ in their propensity to use different plant hosts, a phenotype with clear relevance for identifying crops at risk. However, inconsistencies in the genetic and phenotypic descriptions of these "host strains" has led to controversy about their composition and even existence. In this study, the locus for the Triosephosphate isomerase gene (*Tpi*) is used both as a host strain marker and for phylogenetic analysis. Association of the host choice phenotype with the *Tpi*-derived phylogenetic tree uncovered genetic differentiation between populations that supports the existence of the host strains and provided evidence that they are subject to different selection pressures. This correspondence of differential host use with *Tpi* was demonstrated for populations from a broad geographical range and supports the involvement of one or more *Z*-chromosome functions controlling the phenotype. Comparisons of collections from multiple locations identified significant differences in the efficacy of different molecular markers that implicate regional variations in host strain behavior.

## Introduction

The noctuid moth *Spodoptera frugiperda* (JE Smith) (Lepidoptera: Noctuidae), commonly called the fall armyworm (FAW), is an important agricultural pest native to the Americas that has become a global concern with its 2016 discovery in western Africa followed by infestations detected in India (2018), China (2018–2019), and Australia (2020). The estimated cost to the Eastern Hemisphere is in the billions of USD primarily due to reduced yields in corn and sorghum [1–3]. FAW has been associated with hundreds of plant types in the Americas [4, 5], leaving open the possibility of even higher Eastern Hemisphere losses in crops such as millet, forage grasses, rice, and possibly in native food sources whose susceptibilities to FAW are unknown [6].

Two populations of FAW have been reported in the Western Hemisphere that differ in their distribution on crops of economic importance. This phenotype is the basis for their designation as host strains, with the C-strain typically the majority in corn, sorghum, and cotton

**Data Availability Statement:** All sequence data used for the phylogenetic analyses are available in the NCBI Genbank database. Collections for Tpil4a200 phylogeny: ArgL (Accession numbers ON586157-ON586213), BraL (ON586214 - ON586265), FLL (ON586266 - ON586329), FLT

(ON586330 - ON586429). Collections for TpiE3-4 phylogeny: BraL (ON586430 - ON586501), FLL (ON586502 - ON586558). Tpi sequence for S. eridania (ON586559) and S. littoralis (ON586560). The S. litura TpiE3-4 sequence was obtained from Genbank accession number XM022977018.

**Funding:** The author R.N.N. received support came from the Agricultural Research Service of the United States Department of Agriculture (6036-2200-30-00D) and USAID PASA (908-0210-012). The funders had no role in study design, data collection and analysis, decision to publish, or preparation of the manuscript.

**Competing interests:** The author has declared that no competing interests exist.

infestations while the R-strain predominates in turf, pasture, millet, and alfalfa [7–10]. Strain identification is therefore of practical importance as a means to identify crops at risk in a given area. For example, strain markers in the Triosephosphate isomerase (*Tpi*) gene indicate that only the C-strain is found in significant numbers in Africa and Asia, consistent with observations that FAW damage has been mostly limited to C-strain preferred plant hosts [11–15]. Based on these observations, programs to prevent the introduction and expansion of the R-strain in the Eastern Hemisphere are recommended [6].

However, the host strain model is complicated by difficulties in identifying strains. The two strains are morphologically indistinguishable, restricting identification to molecular methods that include mitochondrial polymorphisms and a small number of sex-linked nuclear markers [10, 16–18]. Because these loci are not believed to determine strain-specific behavior, their effectiveness is dependent on how tightly linked they are to the genes that do. This means that the validity and accuracy of these markers can only be extrapolated from their correspondence with the host use phenotype [10]. Additional complications arise from the failure of multiple feeding studies to demonstrate a consistent physiological reason for the host specificity. The overall findings indicate that both strains can survive and develop on the same plant hosts under controlled conditions, with variable performances on different hosts (reviewed in [19]) and at least one report that both strains develop fastest on corn [20]. This suggests that both strains can use the same set of hosts, with the observed differential distribution in the field resulting from differences in host preference rather than a physiological necessity. This would explain instances when the strains are found on the "wrong" hosts [8, 10, 21], which has elicited challenges about the legitimacy of the host strains as stable, genetically defined populations [21].

Despite these complications, statistically significant correlations between the genetic markers and host plant use has been documented multiple times in both Americas [7–9, 22], and these findings are the basis for the long-standing belief that the host strains represent an example of incipient sympatric speciation being driven by differences in host use [10, 19, 23]. However, recent whole genome attempts to identify genomic differentiation between the strains has had mixed results, with one study finding nuclear SNPs exhibiting strain-specificity [24] while others did not [25, 26]. This again has led to concerns about whether the host strains exist as genetically differentiated populations [26]. Related to this possibility are studies describing reproductive isolation mechanisms for the strains that may be unrelated to host and habitat selection. These include evidence of strain differences in female pheromone composition [27–29], partial reproductive incompatibility between the strains [19, 23, 30, 31], and strain differences in the timing of mating activity during scotophase [24, 32–34]. Such mechanisms either singly or in combination could allow divergence of the strains independent of differential host selection, *i.e.*, two differentiated populations exist but are not defined by host usage and therefore not "host strains". Such a demotion of host specificity to an irrelevant or even artifactual phenotype would represent a fundamental change in the current understanding of FAW biology and invalidate the use of strain markers to assess what crops are at risk of FAW infestation.

These complications are perhaps to be expected if the strains are at an early stage of speciation [35] when reproductive isolation is incomplete and the populations differ by relatively few genes. Under such conditions, cross-hybridization between strains, which has been demonstrated to occur in field populations [10, 36], could compromise the association of these phenotypes with genetic markers. Because of such uncertainties, additional research is needed to assess the legitimacy of the host strain model for FAW and, if confirmed, the accuracy of the available molecular and genetic tools for host strain identification.

In this paper the host strains are reassessed using the *Z*-linked *Tpi* gene, which encodes for a housekeeping enzyme not thought to be directly involved in strain identity. One *Tpi* exon contains single nucleotide polymorphisms (SNPs), including site gTpi183Y, that show a consistent correlation with the differential plant host use exhibited by the strains [37]. This linkage implicates the mapping of functions controlling host use to the *Z*-chromosome. Adjacent to gTpi183Y is an intron sequence (TpiI4a200) with high genetic variability [38, 39] that facilitates phylogenetic analysis. The trees produced from TpiI4a200 were predicted to provide a phylogenetic description of *Tpi* and that of linked traits (such as the host use phenotype), with the potential to provide new information on the genetic differences between the host strains. This was examined by identifying phylogenetic clusters that were differentially associated with host use and testing whether their comparisons produced data consistent with past findings using other methods. The implications of the results to the genetics and evolution of the host strains and the relative efficacy of commonly used strain markers in different populations are discussed.

## Results

An initial analysis was performed using pheromone trap captures from multiple locations and times that should represent a broad cross-section of populations in southern and central Florida (FLT, Fig 1). The TpiI4a200 segment of the *Z*-linked *Tpi* gene was previously shown to be both highly variable and potentially strain specific based on associations with the closely linked

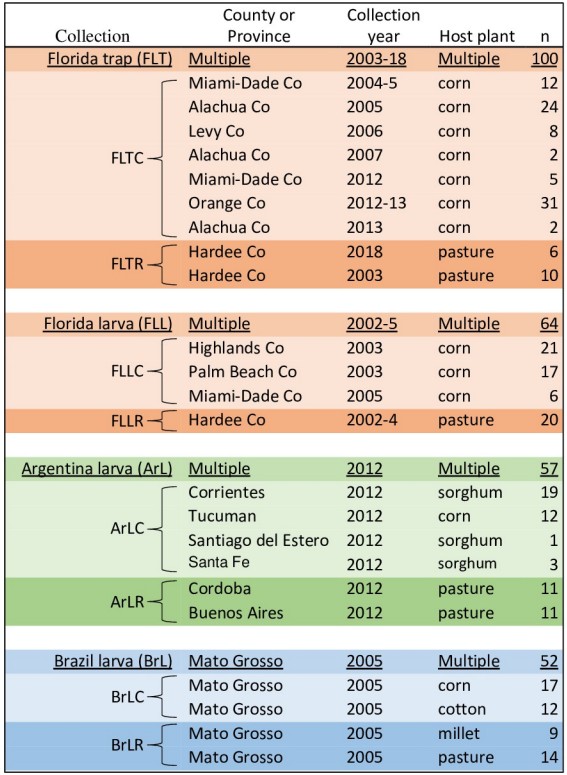
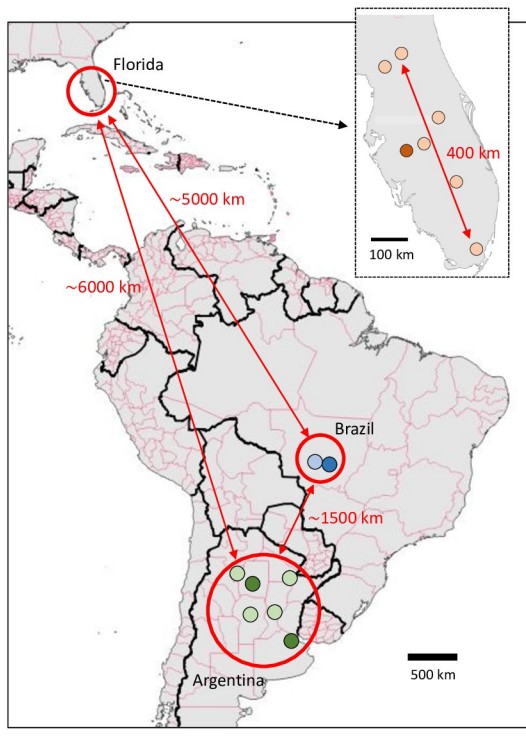

**Fig 1. Source information for specimens used in this study.** Map shows the approximate locations of collections (circles), with orange, green, blue indicating sites in Florida, Argentina, and Brazil respectively. Light tints identify collections from C-strain plant hosts while darker tints show collections from R-strain hosts. The map was created using the Quantum Geographic Information System (version 2.18.2, http://qgis.osgeo.org).

gTpi183Y marker in the adjacent exon TpiE4 (Fig 2) and to commonly used mitochondrial strain markers that map to the Cytochrome oxidase subunit I (*COI*) gene [38, 39]. Markers in these genes identifying the C-strain are designated $C_{COI}$ and $C_{Tpi}$ and those for the R-strain are $R_{COI}$ and $R_{Tpi}$.

## The TpiI4a200 phylogeny

A total of 100 sequences for TpiI4a200 were obtained from the trap collections and these revealed 37 haplotypes. A Neighbor-Joining phylogenetic tree was generated from these sequences and color-coded for whether the plant composition of the collection site was predicted to attract the C-strain ($C_{Host}$ = corn, sorghum, cotton hosts) or R-strain ($R_{Host}$ = pasture and turf grasses, millet, alfalfa) populations (Fig 3A). One phylogenetic group contained 64 specimens where 63 (98%) originated from traps in C-strain habitats. The remaining clusters did not exhibit a clear host preference, with an overall proportion of 58% (21/38) from C-strain trap locations. The same phylogenetic tree was used to assess the correspondence between the phylogenetic groups and the distribution of the *COI* and *Tpi* strain markers. The gTpi183Y site is separated by only 24-bp from the TpiI4a200 intron segment and so a tight association with the TpiI4a200 phylogeny was expected (Fig 2). This was observed as the $C_{Tpi}$ allele diagnostic of the C-strain was found in 100% (64/64) of the phylogenetic group that was predominantly from C-strain habitats compared to only 14% (5/36) of the remaining specimens (Fig 3B). Despite being on an independent genetic element, the mitochondrial *COI* strain marker showed a similar bias with 73% (47/64) of the C-strain habitat group exhibiting $C_{COI}$ compared to 11% (4/36) in the rest of the tree (Fig 3C). From these findings two cluster groups or "clans" were defined with one predominated by metrics associated with the C-strain (ClanC) and the other characterized by a majority $R_{Tpi}$ and $R_{COI}$ composition but with no obvious plant host specificity (ClanR).

## Identification of ClanC and ClanR in all collections

The flight capability of adult FAW makes pheromone trapping only an approximate indicator of the host plant from which the collected specimen originated. A more accurate identification comes from larval surveys where specimens are picked directly from the host. Larvae from multiple locations and host plants in Florida (FLL), Brazil (BrL), and Argentina (ArL) were analyzed with the TpiI4a200 segment. The sequences were pooled and compared to the ClanC and ClanR haplotypes from FLT. The ensemble Neighbor-Joining phylogenetic tree showed that the haplotypes from each collection were widely dispersed with no obvious geographical

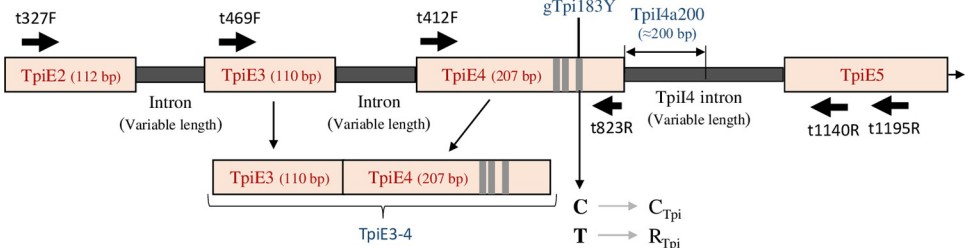

**Fig 2. Diagram of the portion of the FAW *Tpi* gene analyzed in this study.** The approximate location and direction of relevant primers used for PCR amplification and DNA sequencing are denoted by large arrows. The size ranges of exon and intron segments are not drawn to scale. Text in red identifies genetic features used for strain identification or phylogenetic analysis. Vertical grey bands in the gene indicate strain-specific SNP loci including the diagnostic gTpi183Y site that defines the $C_{Tpi}$ and $R_{Tpi}$ haplotypes.

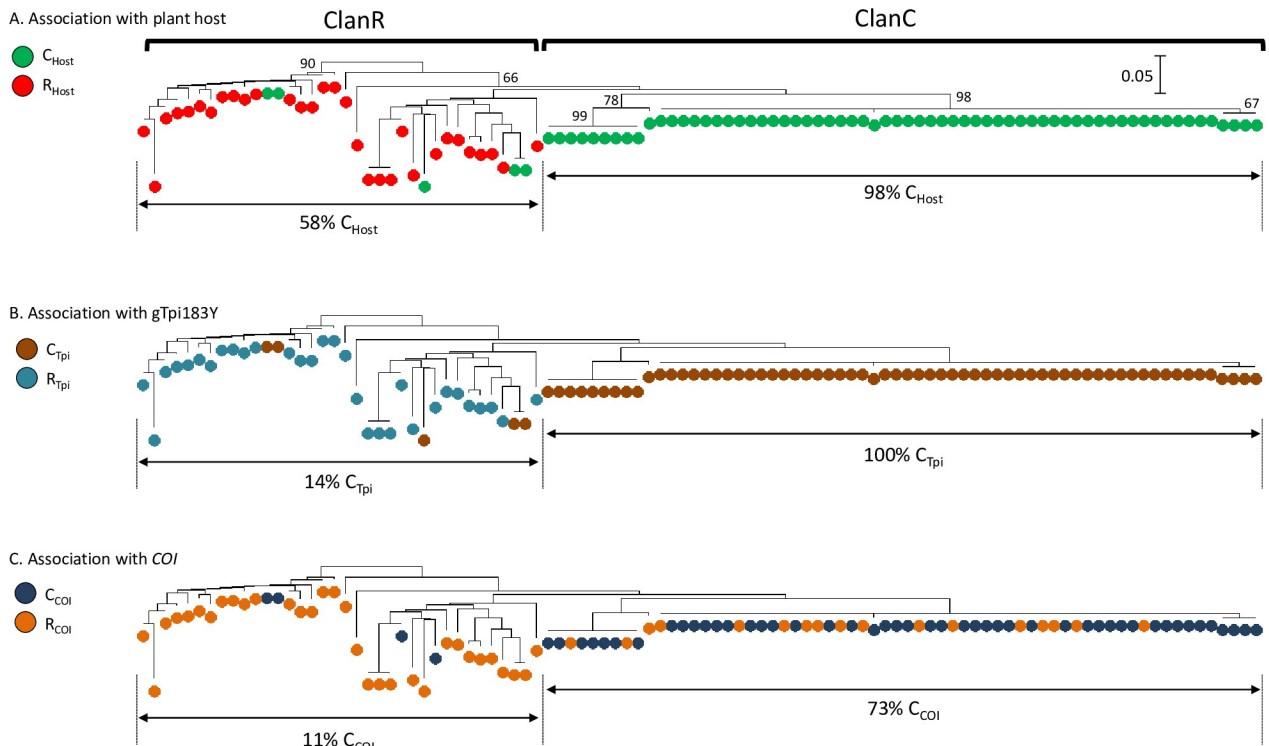

**Fig 3. The association of strain phenotypes with a phylogenetic tree derived from the Florida pheromone trap collection as inferred by the Neighbor-Joining method.** This analysis included 100 sequences of the approximately 200-bp TpiI4a200 segment. A, Specimens are color-coded for the predominant host category of their trap location. $R_{Host}$ describe trap locations attractive to the R-strain and $C_{Host}$ trapping in habitats preferred by the C-strain. Numbers indicate bootstrap values for selected branches. Distance bar represents base substitutions per site. B, Same tree as A but with specimens identified by the gTpi183Y marker. C, Same tree as A but with the *COI* strain identification indicated.

clustering (Fig 4A). The same tree was color-coded showing the locations of the FLT ClanC (blue) and ClanR (brown) haplotypes (Fig 4B). The ClanC haplotypes identified a cluster comprised of 151 sequences that resolved into 15 TpiI4a200 haplotypes. The remaining specimens

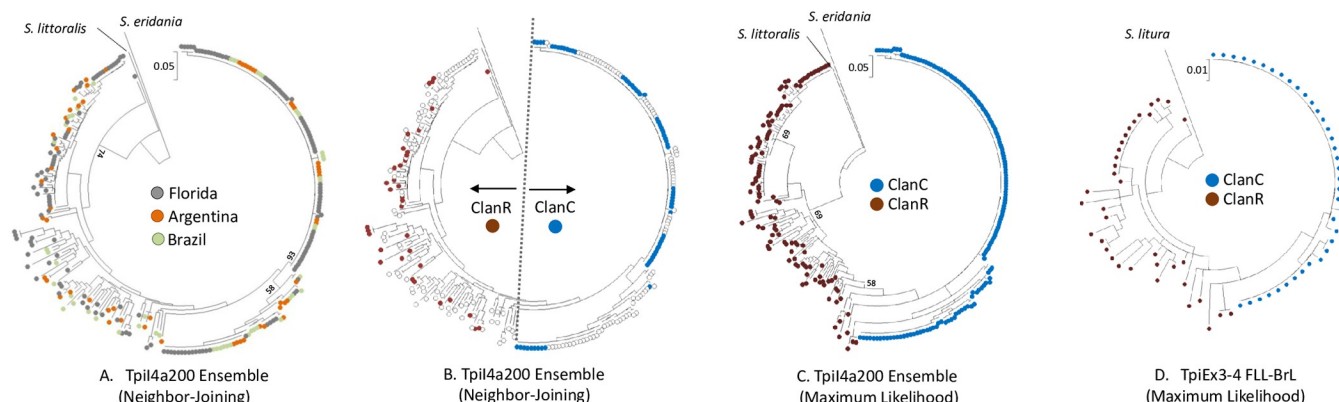

**Fig 4. Comparison of the four collections by phylogenetic analysis.** For each tree, bootstrap values are provided at selected branches. Half bracket with number indicates substitutions per site. A, Neighbor-Joining phylogenetic tree representing the ensemble of all four collections based on the TpiI4a200 segment. Sequences from *S. eridania* and *S. littoralis* are outgroups. Sequences are color-coded by their geographical origin. B, Same phylogenetic tree as in A but with the FLT sequences color-coded for their ClanC or ClanR identity and the remaining sequences indicated by a clear circle. C, Maximum-Likelihood phylogenetic tree using the same ensemble data set as in A. All sequences are color-coded for their ClanC or ClanR identity based on the tree in B. D, Maximum-Likelihood phylogenetic tree for the FLL and BrL collections derived from the TpiE3-4 segment. The *S. litura* sequence is the outgroup root. ClanC and ClanR identity was based on the tree in B.

associated with ClanR was more diverse, with their 122 specimens producing 91 TpiI4a200 sequence variants. Two Spodoptera species were used as outgroups *S. littoralis* (Boisduval) and *S. eridania* (Cramer) and both were rooted within ClanR (Fig 4A and 4B).

To assess the consistency of this cluster pattern the same TpiI4a200 sequence data set was analyzed using Maximum-Likelihood, which is generally believed to provide a more accurate approximation of evolutionary relationships [40] (Fig 4C). The sequences were color-coded for their ClanC/ClanR identity as defined by Fig 4B. The results were comparable to that obtained with Neighbor-Joining showing a segregation of ClanC from ClanR and outgroup rooting localized to the ClanR group.

One concern was that although the outgroup sequences did align with the FAW TpiI4a200 segment, the homology appeared marginal by eye (S1 Fig). To confirm this rooting pattern the *Tpi* exon segments TpiE3-4 (Fig 1) were sequenced from the Florida larval and Brazil collections. The much lower genetic variability in the exon sequences allowed for more reliable alignment of the outgroup, in this case from the *Spodoptera* species *S. litura* (Fabricius) (S2 Fig). The Maximum-Likelihood phylogenetic tree produced by the TpiE3-4 sequences retained the separation of the ClanC and ClanR groups, with the outgroup rooted once again in ClanR (Fig 4D).

Comparisons of the trees revealed substantial differences between ClanC and ClanR with respect to the frequency and geographical distribution of the TpiI4a200 haplotypes. The ClanR group was comprised of 91 haplotypes of which 79 (87%) were private haplotypes (found only once in the surveyed collections), which made up 65% of the 122 ClanR specimens. No ClanR haplotype was common to Florida, Brazil, and Argentina. In contrast, the 151 specimens in ClanC produced only 15 TpiI4a200 haplotypes of which five were private haplotypes (3% of specimens). Unlike ClanR, 125 specimens (83%) in ClanC carried a haplotype found in all three locations, which was due primarily to a single haplotype that made up 61% of ClanC (Fig 4A).

## The host use phenotype identifies two genetically differentiated populations

The TpiI4a200 sequence dataset derived from all four collections was grouped by the association of each sequence with either C-strain ($C_{Host}$) or R-strain ($R_{Host}$) plant hosts. The $C_{Host}$ group exhibited a higher mean haplotype diversity (Hd) than the $R_{Host}$ group, but this was not statistically significant ($t = 1.75$, $df = 6$, $P = 0.131$, Fig 5A). However, a statistically significant difference was observed for nuclear diversity ($\pi$) where in all four collections the $R_{Host}$ $\pi$ was greater than that of the $C_{Host}$ ($t = 3.52$, $df = 6$, $P = 0.013$, Fig 5B).

One of most commonly used measures for comparing the genetic differentiation between populations is Wright's fixation index ($F_{ST}$), which compares the distribution of allele frequencies [41]. The $F_{ST}$ metric has a maximum of one that is approached with increasing differentiation between populations, while a value of 0 indicates panmixia. Calculations were performed for all pairwise combinations of the collections (S1 Table). All comparisons within the plant host groups gave relatively low $F_{ST}$ scores with a mean of 0.1 for $C_{Host}$ vs $C_{Host}$ and 0.06 for $R_{Host}$ vs $R_{Host}$ comparisons (Fig 5C). Both are significantly lower than the mean $F_{ST}$ of 0.35 obtained from comparisons between the $C_{Host}$ and $R_{Host}$ groups (ANOVA: $F = 15.57$, $r^2 = 0.5547$, $P<0.0001$). The overall results indicate that the populations defined by the use of plant hosts associated with each strain are significantly differentiated from each other. This characteristic was found in populations from both American continents and supports the existence of the host strains and their broad distribution in the hemisphere.

The same analyses were performed on the dataset used in Fig 5A–5C, but this time with the sequences categorized by their ClanC and ClanR identification. The results were comparable

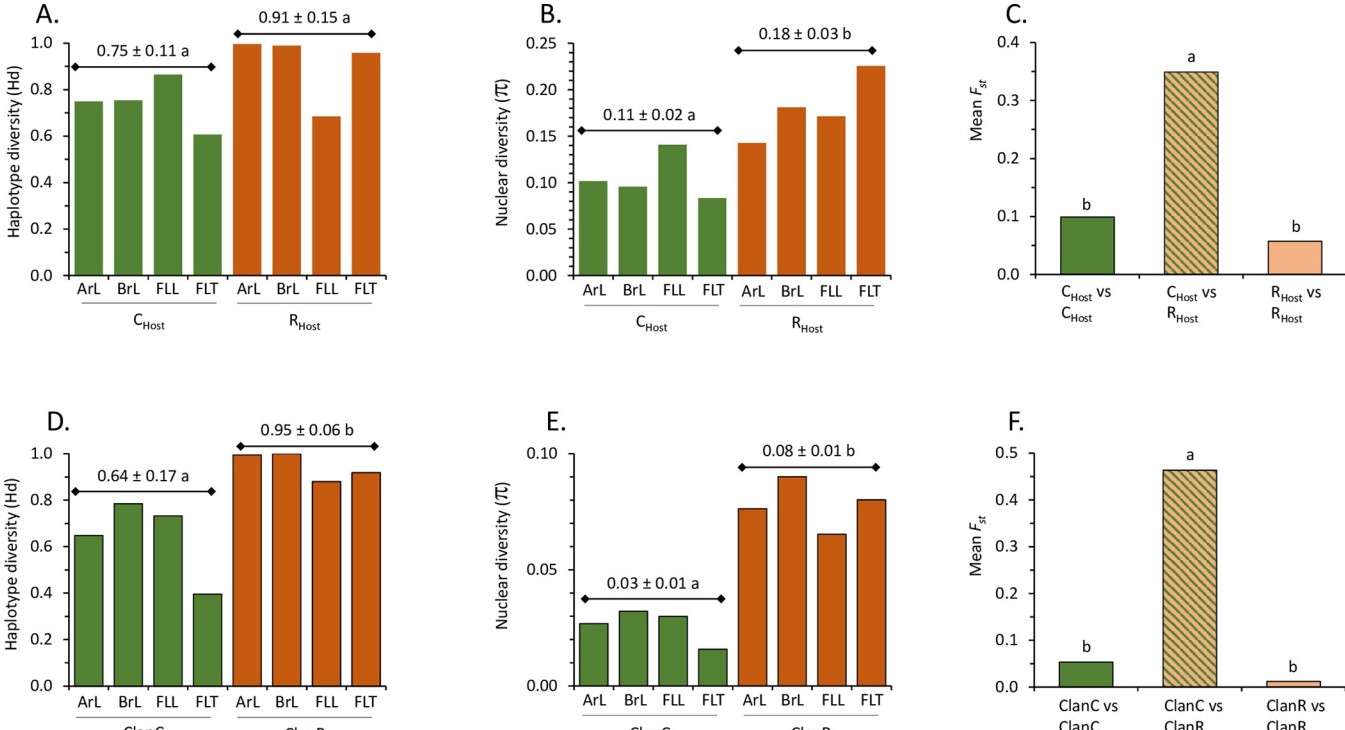

**Fig 5. Measures of genetic variation in population categorized by plant host use or phylogenetic grouping.** A, Haplotype diversity (Hd) for each collection subdivided by plant host ($C_{Host}$ and $R_{Host}$). Means are presented over the columns ± the standard deviation, with different lower-case letters indicating a statistically significant difference. B, Nucleotide diversity ($\pi$) for the same set of collections as A. C, $F_{st}$ values were calculated for all pairwise combinations of $C_{Host}$ and $R_{Host}$ from the individual collections (S1 Table) and the means of within group and between group pairings were analyzed by ANOVA with a multiple comparisons post-test. Different lower-case letters indicate the difference was statistically significant. D, Same as A but grouped by ClanC or ClanR assignment. E, Same as B but grouped by ClanC or ClanR assignment. F, Same as C but for all pairwise combinations of ClanC and ClanR from the collections.

with that observed with the host use phenotype. ClanR showed significantly higher Hd ($t = 3.46$, $df = 6$, $P = 0.013$, Fig 5D) and $\pi$ ($t = 8.24$, $df = 6$, $P = 0.002$, Fig 5E) means than found with ClanC. Similarly, $F_{ST}$ values were significantly higher in comparisons between ClanC and ClanR than comparisons within the groups (ANOVA: $F = 136$, $r^2 = 0.9158$, $P < 0.0001$, S1 Table, Fig 5F). These observations suggest that the genetic differentiation exhibited when the collections were defined by host use was due to their association with ClanC and ClanR.

## Quantifying the association of host strain identifiers with ClanC and ClanR

If the TpiI4a200 clusters are descriptive of the host strains then ClanC should be closely associated with the C-strain characteristics of $C_{Host}$, $C_{Tpi}$, and $C_{COI}$, while ClanR will be linked with the R-strain counterparts and therefore have little correspondence with $C_{Host}$, $C_{Tpi}$, and $C_{COI}$. This should be particularly true for the *Tpi* marker because of the physical linkage of gTpi183Y with the TpiI4a200 segment (Fig 2). Specifically, the expectation was for a high ClanC $C_{Tpi}$ and a low ClanR $C_{Tpi}$ frequency, both of which were observed with the means significantly different (Fig 6A and 6E).

The other C-strain pairings of ClanC $C_{COI}$ and ClanC $C_{Host}$ were found at mean frequencies comparable to that of ClanC $C_{Tpi}$, with the ClanC $C_{COI}$ frequency lowest in all collections but not significantly different (Fig 6A–6C). No significant differences in the means were found for the ClanR pairings as well (Fig 6D–6F). Pair-wise comparisons between the ClanC and

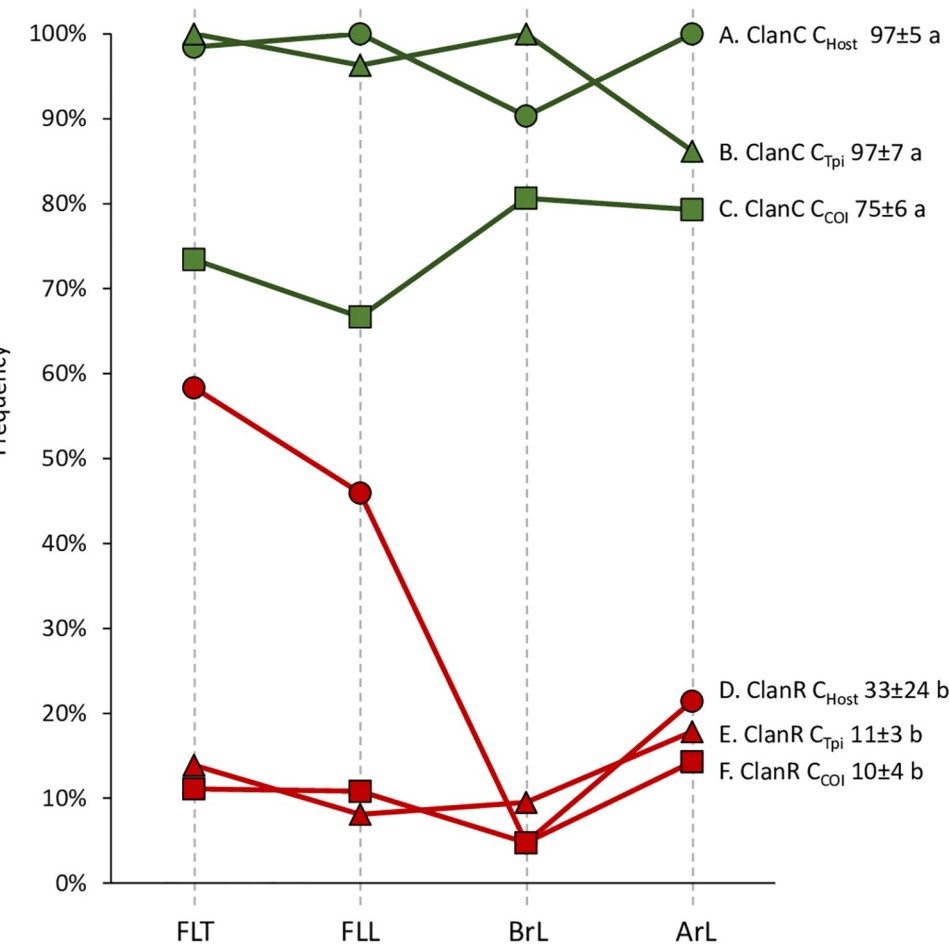

**Fig 6. Comparison of the C-strain metrics $C_{COI}$, $C_{Tpi}$, and $C_{Host}$ for ClanC and ClanR from each collection.** Graph compares the frequencies of each C-strain phenotype in the ClanC (green) and ClanR (red) groups. Mean frequencies ± standard deviation are shown next to curve labels. Means that are significantly different ($P<0.05$) by two-tailed $t$-test analysis have different lower-case letters.

ClanR curves reveal statistically significant differences between the groups and confirm a correspondence between ClanC and ClanR with genetic and behavioral phenotypes indicative of the C-strain and R-strain, respectively. This was particularly the case in the South American collections where the association of the phylogenetic clusters with the host phenotype was comparable with that found for the genetic markers.

The ClanR $C_{Host}$ curve showed a large variation in $C_{Host}$ frequency where about half of the ClanR in Florida (FLT, FLL) was associated with $C_{Host}$ compared to only 5% (1/20) and 21% (6/28) in the Brazil and Argentina collections, respectively (Fig 6D). While this difference was not detected in the statistical comparison of the means it was seen using *chi*-square analysis where the ClanR $C_{Host}$ frequency distribution was significantly different from the frequencies of ClanR with either the $C_{COI}$ or $C_{Tpi}$ markers (Table 1). These results suggest that the ClanR population exhibits a higher level of specificity for R-strain plant hosts in Argentina and Brazil than it does in Florida.

**Table 1. *Chi*-square analysis of C-strain metrics in ClanC and ClanR.**

| Comparisons | $Chi^2$ | df | P-value |
|---|---|---|---|
| ClanC $C_{host}$ vs ClanC $C_{COI}$ | 2.53 | 3 | 0.47 |
| ClanC $C_{COI}$ vs ClanC $C_{Tpi}$ | 3.86 | 3 | 0.28 |
| ClanC $C_{Host}$ vs ClanC $C_{Tpi}$ | 1.64 | 3 | 0.65 |
| ClanR $C_{Host}$ vs ClanR $C_{COI}$ | 9.05 | 3 | 0.03* |
| ClanR $C_{COI}$ vs ClanR $C_{Tpi}$ | 2.78 | 3 | 0.43 |
| ClanR $C_{Host}$ vs ClanR $C_{Tpi}$ | 19.64 | 3 | 0.0002* |

## Evidence that ClanC and ClanR are evolving under different selection pressures

If ClanC and ClanR provide a phylogenetic description of the host strains, then the comparison of these clusters can provide insight into how the strains are diverging. The ClanC and ClanR groups were analyzed by two metrics commonly used to distinguish between neutral and selective evolution. Tajima's *D* is based on comparisons between the number of segregating sites and the mean number of nucleotide differences [42], while the Fu and Li's *F*\* metric is calculated from the number of singletons relative to the average nucleotide differences between the compared pair of sequences [43]. In both cases the null hypothesis is neutral evolution (no selection), which gives a value of 0, while positive values indicate fewer than expected rare alleles and negative values an overabundance of these.

   None of the regional collections individually showed a statistically significant departure from the null hypothesis. However, there was a consistent difference in the signs of the metrics, with both *D* and *F*\* being positive for all locations in ClanC and negative in ClanR (Fig 7). The differences between the means were statistically significant for both D ($t = 5.54$, $df = 3$, $P = 0.012$) and F\* ($t = 12.46$, $df = 3$, $P = 0.0011$). The same trend was seen when the collection sequences were pooled to give ensemble D and F\* values that were positive for ClanC and

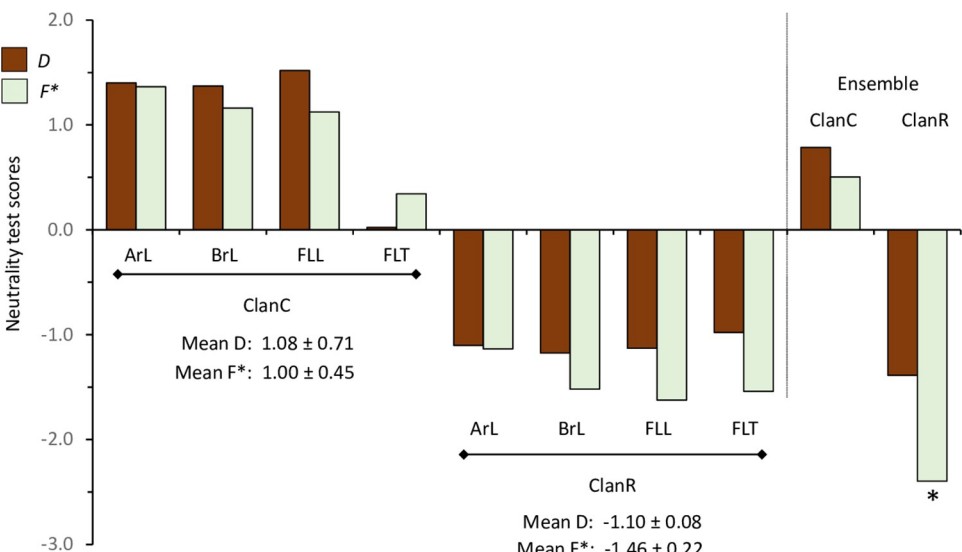

**Fig 7. Neutrality test results for Tajima's *D* and Fu and Li's *F*\* calculations for each collection according to their ClanC and ClanR groupings.** Asterisks under columns indicate a statistically significant departure from the null hypothesis of neutral evolution. In the Ensemble groupings the collections from all locations are pooled.

negative for ClanR, with F* for the latter showing a significant difference from the null hypothesis. Overall, these data indicate that ClanC and ClanR are experiencing different selection pressures, consistent with the high $F_{ST}$ values found in comparisons between ClanC and ClanR.

## Discussion

Host strains, or host races, are defined as being primarily distinguished by their differential use of hosts and are presumed to represent an intermediate stage of sympatric speciation with appreciable gene flow (to distinguish them from host-associated species) and detectable genetic differentiation [44]. FAW has long been proposed to represent an example of host strains, with two populations with different propensities to use certain plant hosts that can be distinguished by a small set of mitochondrial and nuclear genetic markers. However, this proposal has been challenged by the failure of recent whole genome SNP studies to consistently find genetic differences between presumed host strain populations [25, 26].

In this study, phylogenetic analysis of this TpiI4a200 intron segment using collections from three geographically distant locations consistently identified genetically related clusters (ClanC and ClanR) that corresponded to the strain-specific plant host phenotype and thereby were assumed to provide a genetic description of the host strains. This assertion is supported by comparisons between ClanC and ClanR that agree with past host strain findings based on other methods. For example, ClanR exhibits significantly higher haplotype diversity and three-fold higher nucleotide diversity than ClanC (Fig 7A and 7B). This finding is consistent with AFLP (Amplified Fragment Length Polymorphism) studies reporting higher genetic variation in the R-strain than C-strain [21]. In addition, the AFLP results combined with behavioral observations led to the conclusion that the R-strain is the ancestral population from which the C-strain is derived [45]. This agrees with the TpiI4a200 and TpiE3-4 phylogenetic analyses that indicated by outgroup rooting the ancestral nature of ClanR, with ClanC emerging from a distant ClanR branch (Fig 4).

An essential characteristic of host strains is that they should exhibit genetic differentiation that corresponds to differential host use. This was observed for FAW by $F_{st}$ analysis of the Tpi4a200 sequence database categorized by $C_{Host}$ and $R_{Host}$ groups (Fig 5C). The similar result for comparisons between ClanC and ClanR (Fig 5F) underscores the correspondence of the host use phenotype with the phylogenetic groups. Based on this relationship, the finding that ClanC and ClanR exhibit very different Tajima's *D* and Fu and Li's *F** metrics (Fig 7) suggests that the strains are experiencing substantially different selection pressures that could be related to their differential host use. These observations support the existence of the host strains as genetically diverging populations and the legitimacy of the ClanC and ClanR groups as descriptors of the strains.

The $F_{ST}$ analyses are especially striking when viewed in the context of the spatial and temporal separation of the collections. The Argentina collections are over 1500 km and seven years separate from the Brazil collections, and both are over 5000 km distant from the Florida collections (Fig 1). Yet no evidence of significant genetic differentiation was found in comparisons within either the ClanC or ClanR groups from different locations. In contrast, substantial differences were observed in all comparisons between groups (ClanC vs ClanR), even when the collections are contemporaneous and from the same region, with $C_{Host}$ and $R_{Host}$ collection sites typically within 100 km separation. These observations indicate sufficient movement of FAW across the hemisphere to limit differentiation between geographically distant populations of the same strain, while exhibiting genetic isolation between strains even when sympatric in their distribution. The latter indicates that despite evidence of cross-hybridization

between strains in the field, the frequency is not sufficient to compromise genetic differentiation between strains as measured by $F_{ST}$.

One caveat to this set of conclusions is that it is based solely on the Z-linked *Tpi* gene and so may not adequately represent the phylogenetic history of the strains, particularly if autosomal functions are important to strain identity. However, this should not be a significant issue for the host use phenotype since its linkage to *Tpi* indicates mapping to functions on the Z-chromosome (Fig 6A and 6D). This does not preclude the involvement of autosomal functions controlling other phenotypes that may be involved in differentiating the strains (and there is genetic evidence for at least two such loci [24, 46]). But if autosomal functions did have a predominant role in defining host use, then hybridization between strains (which has been demonstrated to occur at significant levels in the field [36]) will substantially compromise its association with *Tpi*. This has not occurred as the *Tpi*-host phenotype linkage can be demonstrated in populations from multiple locations and times in the Western Hemisphere [8, 22, 47]. Additional support for the involvement of the Z-chromosome in host strain identity comes from observations that partial mating incompatibility between the strains is Z-linked [30], and the identification of strain-specific SNPs of which 93% with the highest specificity mapped to the Z-chromosome [24]. If strain identity is predominantly controlled by Z-linked genes, then the inconsistencies in finding strain-specific SNPs in recent studies [24–26] could be a technical issue, reflect variability in how much the Z-chromosome (which is one of 31 chromosomes in FAW) was represented.

The correspondence of the ClanC and ClanR clusters with plant hosts is not absolute, with an average of 3% of ClanC discordantly associated with R-strain hosts and 33% of the ClanR population from C-strain host (Fig 5). Such variability is not surprising if the host strains are at an intermediate stage of sympatric speciation [10, 19, 23], such that the penetrance of the distinguishing phenotypes or the degree of reproductive isolation are incomplete. Nevertheless, the results here demonstrated that the host use phenotype is sufficiently consistent to be observed in populations from multiple locations and time periods, supporting it being a fundamental phenotype differentiating the strains. This does not preclude the involvement of other processes, particularly those that contribute to reproductive isolation through assortative mating or hybrid incompatibility as being part of the host strain phenotype [29, 48, 49]. However, at this time differential host use is the only trait demonstrated in host strain populations from both American continents.

The correspondence of the ClanC and ClanR clusters to the host phenotype justify their use as a proxy for the C-strain and R-strain, respectively, and facilitated new insights into the characteristics of the strains. The ClanR groups from Brazil and Argentina displayed a strong association with R-strain host plants and levels comparable to the correspondence of ClanC with C-strain hosts (Fig 7). However, no host bias was observed for ClanR in the Florida collections suggesting that this population is behaving as a generalist with respect to host choice. If confirmed, this would represent the first example of geographical differences in strain behavior. While speculative, it is possible that the Florida behavior is a consequence of differences in climate and host availability between the two Americas. The freezing winters experienced in most of North America means that FAW must winter in relatively small areas in southern Florida, southern Texas, and Mexico, where conditions are more temperate and where corn is a common winter crop and often the predominant FAW host. This is followed by a rapid northward migration as far north as Canada in the spring and summer months [4] that correlates with the northward progression of corn plantings, which provides an abundant and perhaps essential early season host for high density migratory populations [50, 51]. In comparison, the climate of much of South America is compatible with FAW year-round, causing much less seasonal variability in host availability for both strains [52]. It is plausible that

these conditions may create selection pressures unique to North America favoring the adaptation of the R-strain to corn.

In summary, the results presented support the existence of the FAW host strains defined as genetically differentiated populations that differ in their propensity to use different host plants. Evidence for such populations were found at collections sites in both American continents, a broad distribution consistent with it being a general characteristic of the species. The phylogenetic analysis of the TpiI4a200 intron segment and the association of clusters to the host strains provided new insight into the genetics and behavior of the strains. These include indications that the strains may differ in their host specificity depending on location, quantification of differences in the selection pressure driving strain divergence, and additional support for the importance of Z-linked functions in determining the differential preference for plant hosts.

## Materials and methods

### Source and treatment of specimens

The collections used in the phylogenetic analysis are described in Fig 1. All specimens are from field collections described in earlier studies and fall into four groups, Florida trap collections [22, 36–39, 53–55] and Florida larvae [22, 56, 57] were done by R. Meagher. Argentina larval collections were coordinated by G. Murúa [8], and Brazil larvae were obtained from P. Silvie [22, 58]. The larvae were collected directly from their host with only one specimen taken per plant to reduce the frequency of siblings. Upon collection, the specimens were dried then stored refrigerated or frozen until transport by mail to CMAVE, Gainesville, FL USA for DNA preparation. Nuclear and mitochondrial DNA were isolated by methods previously described using spin-column chromatography according to manufacturer's instructions (Zymo Research, Orange, CA) for final purification and concentration [38].

### DNA sequence analysis

Segments from the *COI* and *Tpi* genes were amplified by polymerase chain reaction (PCR) amplification in a 30-μl reaction mix containing 3 μl of 10X manufacturer's reaction buffer, 1 μl 10mM dNTP, 0.5 μl 20-μM primer mix, 1 μl DNA template (between 0.05–0.5 μg), 0.5 units Taq DNA polymerase (New England Biolabs, Beverly, MA) with the remaining volume water. The thermocycling program was 94˚C (1 min), followed by 33 cycles of 92˚C (30 s), 56˚C (45 s), 72˚C (45 s), and a final segment of 72˚C for 3 min. The *COI* strain marker segment was amplified using primers c891F (5′-TACACGAGCATATTTTACATC-3′) and c1303R (5′-CAGGA TAATCAGAATATCGACG-3′) with sequencing using c891F. Analysis of the gTpi183Y SNP and the sequences of TpiI4a200 was obtained by PCR amplification using t412F (5′-CCGGA CTGAAGGTTATCGCTTG-3′) and t1140R (5′-GCGGAAGCATTCGCTGACAACC-3′) or t1195R (5′-AGTCACTGACCCACCATACTG-3′) followed by DNA sequencing using t412F or t1140R. The sequence for TpiE3-4 was obtained in two steps. The sequence for TpiE3 was came from the PCR amplification using t327F (5′-CGCACAAAACTGCTGGAAG-3′) and t823gR (5′-GTCTTTCCGGTGCCAATAG-3′) and sequencing using t327F. The sequence for TpiE4 was from the sequencing of the amplified product from t469F (5′-AAGGACATCGGAGCCAA CTG-3′) and t1140R. The sequence data was combined to generate TpiE3-4.

The TpiI4a200 sequences from the sibling species *S. littoralis* and *S. eridiania* were obtained using a variation of nested PCR. Genomic DNA from these species was previously described and characterized [59]. The first PCR amplification step was performed as described above but with a 48˚C annealing temperature and with primers t469F and t1195R. The completed reaction was diluted 1:2 and 1 μl of the mix re-amplified with a 48˚C annealing temperature and with primers t412F and t1140R. All primers were synthesized by Integrated DNA Technologies (Coralville, IA).

The PCR products were separated by agarose gel electrophoresis and purified using the Zymoclean Gel DNA Recovery Kit (Zymo Research, Orange, CA). The isolated fragments were directly analyzed by DNA sequencing performed by Azenta Life Sciences (Chelmsford, MA). DNA sequence analysis including alignments were performed using Geneious Prime 2021.1.1 (Biomatters, Auckland, New Zealand).

### Analysis of genetic variation between populations

Quantification and comparisons of genetic variability within and between populations were performed using the DNAsp software package [60]. This include calculations of haplotype diversity (Hd) and nucleotide diversity (p), Tajima's $D$ [42] and Fu and Li's $F^*$ [43] analyses, and Wright's fixation index ($F_{ST}$) [41]. Statistical analyses were conducted using GraphPad Prism version 7.00 for Mac (GraphPad Software, La Jolla California USA). Analyses performed included two-tailed, unpaired $t$-tests, *chi*-square, and ordinary one-way ANOVA with multiple comparisons post-test. Generation of graphs were done using Excel and Powerpoint (Microsoft, Redmond, WA). Sequences used in the phylogenetic analysis have been deposited into GenBank.

Phylogenetic analyses were performed on the TpiI4a200 and TpiEx3-4 segments (Fig 2). Sequences from the collections were aligned and edited as necessary for the particular segment using Geneious Prime 2021.1.1 software [61]. Neighbor-Joining analysis [62] was performed using both Mega version 11 [63] and Geneious Prime 2021.1.1. Maximum Likelihood [64] trees were constructed using Mega version 11. All analyses underwent standard bootstrap with 100 pseudo replicates. Smaller trees are presented in linear format while larger trees are depicted in a circular format. The evolutionary distances were computed using the Maximum Composite Likelihood method [65]. Unless otherwise noted the trees are diagrammed based on midpoint rooting with the evolutionary analysis conducted by Mega version 11 [63].

### Identification of strain using *COI* and *Tpi*

The C-strain and R-strain are denoted by a C and R, respectively with a subscript specifying how the identification was made. Strain identification using *COI* was done as described previously [38]. Briefly, a SNP designated mCOI1164D is the diagnostic marker with an A or G at the site indicating $C_{COI}$ and a T identifying $R_{COI}$. Strain identification with *Tpi* used polymorphisms in the fourth exon (TpiE4) of the presumptive coding region with the SNP designated gTpi183Y generally considered the diagnostic marker to identify $C_{Tpi}$ or $R_{Tpi}$ (Fig 2).

A complication of *Tpi* is that male specimens carry two copies of the *Z*-linked *Tpi* gene and so can be heterozygous for either gTpi183Y or TpiI4a200. Because DNA sequencing is performed on the PCR product amplified directly from the genomic DNA preparation, heterozygosity for polymorphisms can produce overlapping information. Such ambiguous sequences were not included in the analysis. The frequency of these events varied by collection and was particularly in the pheromone trap population since these are all males. The larval collections were not identified by sex but should be comprised of both genders. The total number of specimens sequenced from each collection to obtain the unambiguous sequence data used in this study were 94 for the ArL collection, 73 for BrL, 87 for FLL, and 524 for FLT.

### Supporting information

**S1 Fig. Alignment of the TpiE4 and TpiI4a200 segments from FAW, *S. eridania*, and *S. littoralis*.**
(TIF)

**S2 Fig. Alignment of the third and fourth coding exons from the *Tpi* gene from FAW and *S. litura*.** The *S. litura* sequence was obtained from Genbank accession number XM022977018.
(TIF)

**S1 Table. Calculations of $F_{ST}$ for populations categorized by host use ($C_{Host}$, $R_{Host}$) or phylogenetic groups (ClanC, ClanR).** ClanC or $C_{Host}$ are designated C while ClanR or $R_{Host}$ are designated R.
(DOCX)

## Acknowledgments

Dr. J.M.G. Thomas is recognized for technical assistance in preparing the specimens and for comments about the manuscript. Thanks to Dr. Greg Sword and Dr. Robert Meagher for helpful comments and suggestions. The use of trade, firm, or corporation names in this publication is for the information and convenience of the reader. Such use does not constitute an official endorsement or approval by the United States Department of Agriculture or the Agricultural Research Service of any product or service to the exclusion of others that may be suitable.

## Author Contributions

**Conceptualization:** Rodney N. Nagoshi.

**Data curation:** Rodney N. Nagoshi.

**Formal analysis:** Rodney N. Nagoshi.

**Funding acquisition:** Rodney N. Nagoshi.

**Investigation:** Rodney N. Nagoshi.

**Methodology:** Rodney N. Nagoshi.

**Project administration:** Rodney N. Nagoshi.

**Resources:** Rodney N. Nagoshi.

**Validation:** Rodney N. Nagoshi.

**Visualization:** Rodney N. Nagoshi.

**Writing – original draft:** Rodney N. Nagoshi.

**Writing – review & editing:** Rodney N. Nagoshi.

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
