## [Decision Letter · Decision Letter 0]

23 Aug 2022

PONE-D-22-19114The phylogeny of host choice in fall armyworm, a global agricultural pest, confirms the fundamental role of the Z-chromosome in strain divergence.PLOS ONE

Dear Dr. Nagoshi,

Thank you for submitting your manuscript to PLOS ONE. After careful consideration, we feel that it has merit but does not fully meet PLOS ONE’s publication criteria as it currently stands. Therefore, we invite you to submit a revised version of the manuscript that addresses the points raised during the review process.

Both reviewers who critically read this manuscript felt that it requires major revisions to meet publication standards. Therefore, I request you to carefully consider the suggestions and critiques of the reviewers to improve the scientific  robustness of this manuscript.  Especially, the concerns raised by the reviewer 2 about the conclusions on the dominant role of the Z-chromosome needs to be address as this is closely related to the title of the manuscript. As you may be aware, sequence data from a single gene may not be extrapolated to draw conclusions on the divergence of an entire chromosome. 

We look forward to receiving your revised manuscript.

Kind regards,

Omaththage P. Perera, Ph.D.

Academic Editor

PLOS ONE

Journal Requirements:

"The author R.N.N. received support came from the Agricultural Research Service of the United States Department of Agriculture (6036-2200-30-00D) and USAID PASA (908-0210-012)."

Reviewers' comments:

Reviewer's Responses to Questions

**Comments to the Author**

1. Is the manuscript technically sound, and do the data support the conclusions?

Reviewer #1: Partly

Reviewer #2: Partly

2. Has the statistical analysis been performed appropriately and rigorously? 

Reviewer #1: No

Reviewer #2: N/A

3. Have the authors made all data underlying the findings in their manuscript fully available?

Reviewer #1: Yes

Reviewer #2: Yes

4. Is the manuscript presented in an intelligible fashion and written in standard English?

Reviewer #1: Yes

Reviewer #2: Yes

5. Review Comments to the Author

Reviewer #1: The research is solid but in part, there are significant problems between the objective of the work, part of the methodology used for this purpose and the interpretation of the data.

This work seeks to reflect the existence of both strains and their association with host plants from a phylogenetic approach. However, from a conceptual and interpretive point of view, how this approach is proposed is not entirely correct.

The author seeks “the detection of clades expressing the strain specific host phenotype, thereby providing a cladistic definition of the host strains based on common ancestry”

Conceptually, there are a number of points that need to be clarified. First, phylogenetic reconstructions based on common ancentrality are inferred using cladistics, a methodology that provides phylogenetic hypotheses (not a cladistic definition) that assess the genealogical/evolutionary relationships of a given taxonomic group. Second, cladistics analyzes phylogenetic relationships by character based methods, not through distance methods such as N-J. Conceptually, the terminology in some points of the manuscript must be correctly stated.

Although phylogenetic relationships can be inferred by analyzes such as the distance based methods used here (which serve as a first approximation in most cases), if more robust and informative results are desired, it is highly recommended that the phylogeny of the host choice in FAW is performed by other methods.

The first results and conclusions that the author shows are the delimitation of two clades defined according to the host plant from which the individuals were collected (traps/larvae) and the identification by molecular markers. This does not seem phylogenetically correct; the delimitation is very simplistic. The trees in figures 3 and 4 show 2 large, well-defined groupings. One of them has 2 subgroups, one with most of the individuals associated with C-host and the other, a sister group to the one just described, with individuals from both hosts. The other grouping is also constituted by individuals from both hosts. The groupings do not appear to have a phylogenetic (or even genetic) basis associated with the hosts. The same trend was observed in the analysis of phylogenetic comparison between regions (Figure 6), with the formation of several groupings.

Therefore, the two clades proposed by the author is artificial without a phylogenetic support. The main concern is that conclusions are made based on this. The trees do not show resolved groupings related to strains and their association with the host plant.

In conclusion I think that the entire study would benefit with a reworking of context, should be rethought, conceptually and methodologically, based on the objective pursued. S. frugiperda is a pest of enormous economic importance, but to what extent both strains are associated with host plants is not clear. Studies from different approach are necessary to propose control and mitigation strategies for this species.

Reviewer #2: In the current study, the author collected fall armyworms from Florida, Brazil and Argentina and then sequenced an intron of the Tpi gene to develop phylogenies that illustrate relationships between strains across these three geographic regions. They then used known genetic markers to show the relationship between strains on these phylogenies, and to compare that to the host plant near which each individual was collected. The study is well written and the data is interesting, but I have a few concerns that should be addressed prior to publication.

First, some of the conclusions drawn in this manuscript are not entirely supported by the data presented. For example, the title of this study comes from the conclusion on lines 337-338 that “this study confirms the dominant role of the Z-chromosome in strain divergence.” However, without looking broadly across the full genome to show that the Z-chromosome played a greater role in strain divergence than the autosomes, this statement cannot be supported. Given that data from only a single gene is presented here, this statement is an overreach that should be removed from the manuscript, and the title should be reconsidered.

Another concern is that the strain specific marker used through much of this study (Tpi exon) is closely linked to the region used to make the phylogenetic trees (Tpi intron). Therefore, the authors should be cautious about the conclusions drawn regarding strain relationships. For example, on lines 167-169, the authors state that the strain marker used on the Tpi exon is more reliable than the COI strain maker, because it more accurately matches their phylogeny. The support provided for this statement is good, however, this relationship could also simply be due to linkage. Based on the data, I don’t think this can be ruled out. The limitations of using these linked genomic regions should be explicitly stated in the discussion.

Finally, the current methods are not sufficient for publication. It is difficult to ascertain which genomic region was used in each analysis, and several population genomic analyses that are described in the results are not mentioned in the methods. The authors should add a section to the methods about their population genetic analyses, being sure to describe which programs were used and any necessary parameters. A few line-specific suggestions are listed below:

Title: Since only a single gene is assessed in this study, there is not enough data here to “confirm the fundamental role of the Z-chromosome in strain divergence.” The authors should change the title to highlight a better supported conclusion.

Line 167-169: The phylogeny produced is not unbiased, but rather is linked to one of the markers being discussed. While the authors explanation is a valid alternative, linkage cannot be ruled out as driving this pattern.

Line 229: Remove the repeated word “up.”

Lines 255-279: These analyses are not described in the methods. Please add a section to the methodology to explain how these statistical tests were performed, which programs and parameters were used. Also, how were individuals assigned to groups in CS vs. CS and RS vs RS comparisons. Were these groups geographic or random?

Line 273-274: How was it determined that the differences in signs for D and F* metrics were statistically significant? This should be added to the methods and justified.

Line 290-291: The authors have not sufficiently shown that host choice is “the defining strain phenotype.” They can state this is an important strain phenotype, but the data seems to show that this phenotype is not strong enough to prevent inter-strain hybridization in C-strain habitats, especially in Florida.

Line 303-311: While the authors can highlight these similarities, I recommend they also mention that their results are based on a single gene.

Line 337-349: As discussed above, this conclusion is not supported by the data presented. The authors should remove this paragraph from their discussion.

Line 356-357: The data presented in this study does not support that host preference plays a “fundamental role” in strain divergence. Rather this study highlights a pattern, that can be ambiguous in some habitats and regions. Consider rewording this section.

Lines 424-434: The authors only describe one method that was used in phylogeny creation. The methods used to create figures 4 and 6 need to be detailed. Also, it’s not entirely clear which segment of the gene was used for phylogeny creation. From the results, it seems like only TpiI4a200 was used, but there don’t seem to be primers listed specific to this region. Please clarify.

Line 420-421: How many heterozygotes were removed and where were they found?

Figure 3: Verify that all labels are correct in this figure. If I’m not mistaken, the CS and RS clade markers are reversed.

6. PLOS authors have the option to publish the peer review history of their article (what does this mean?). If published, this will include your full peer review and any attached files.

Reviewer #1: No

Reviewer #2: No

---

## [Author Response · Author response to Decision Letter 0]

20 Sep 2022

I thank the reviewers for their thoughtful comments and have tried to accommodate each one. This required a nearly complete rewrite of the Introduction and Discussion sections as well as a reorganization of the Results. There has been no change in the data in the results section other than a bit of additional analysis (Table 1) and modifications designed to improve the clarity of the figures. A detailed response to each reviewer is given below with reviewer’s comments in red and bracketed (<<< >>>).

Response to the Academic Editor:

<<< Especially, the concerns raised by the reviewer 2 about the conclusions on the dominant role of the Z-chromosome needs to be address as this is closely related to the title of the manuscript. As you may be aware, sequence data from a single gene may not be extrapolated to draw conclusions on the divergence of an entire chromosome. >>>

The manuscript was substantially rewritten with respect to the Z-chromosome and the stated concern directly addressed in lines 363-380. The sequence data from Tpi is not being used to describe the divergence of an entire chromosome. It is rather used as a tool to identify genetic differences between the populations that use different hosts, a phenotype that past studies have shown is linked to the Tpi gene.

Response to Journal Requirements:

<<<Please include this amended Role of Funder statement in your cover letter; we will change the online submission form on your behalf.>>>

Done.

<<< We note that Figure 1 in your submission contain [map/satellite] images which may be copyrighted…>>>

Noted that map was made using QGIS (not copyright) (lines 112-113).

Response to Reviewer 1:

<<<Reviewer #1: The research is solid but in part, there are significant problems between the objective of the work, part of the methodology used for this purpose and the interpretation of the data.

This work seeks to reflect the existence of both strains and their association with host plants from a phylogenetic approach. However, from a conceptual and interpretive point of view, how this approach is proposed is not entirely correct.>>>

I have substantially rewritten the introduction and discussion and reorganized the results section to more clearly present the logic of the experiments and the justification of the conclusions.

<<<The author seeks “the detection of clades expressing the strain specific host phenotype, thereby providing a cladistic definition of the host strains based on common ancestry”

Conceptually, there are a number of points that need to be clarified. First, phylogenetic reconstructions based on common ancentrality are inferred using cladistics, a methodology that provides phylogenetic hypotheses (not a cladistic definition) that assess the genealogical/evolutionary relationships of a given taxonomic group. Second, cladistics analyzes phylogenetic relationships by character based methods, not through distance methods such as N-J. Conceptually, the terminology in some points of the manuscript must be correctly stated.>>>

The objective of finding a “cladistic definition” has been removed as has all mention of clades. The use of phylogenetic analysis was limited to identifying clusters of genetically related sequences that could be associated with different strain phenotypes. This is described in lines 81-94. 

<<<Although phylogenetic relationships can be inferred by analyzes such as the distance based methods used here (which serve as a first approximation in most cases), if more robust and informative results are desired, it is highly recommended that the phylogeny of the host choice in FAW is performed by other methods.>>>

Phylogenetic trees from Maximum-Likelihood were produced to confirm the N-J tree data (Fig. 4C, D, lines 181-194).

<<<The first results and conclusions that the author shows are the delimitation of two clades defined according to the host plant from which the individuals were collected (traps/larvae) and the identification by molecular markers. This does not seem phylogenetically correct; the delimitation is very simplistic. The trees in figures 3 and 4 show 2 large, well-defined groupings. One of them has 2 subgroups, one with most of the individuals associated with C-host and the other, a sister group to the one just described, with individuals from both hosts. The other grouping is also constituted by individuals from both hosts. The groupings do not appear to have a phylogenetic (or even genetic) basis associated with the hosts. The same trend was observed in the analysis of phylogenetic comparison between regions (Figure 6), with the formation of several groupings. Therefore, the two clades proposed by the author is artificial without a phylogenetic support. The main concern is that conclusions are made based on this.>>> 

The process by which the phylogenetic clusters (now called ClanC and ClanR) were determined is more specifically described (140-142; 162-166) to reduce the “artificiality”. This involves pooled (ensemble) phylogenetic analysis using both Neighbor-Joining and Maximum-Likelihood analysis to show that the ClanC and ClanR from each location are phylogenetically related and consistently observed in the surveyed locations (Fig4B, 181-186).

<<<The trees do not show resolved groupings related to strains and their association with the host plant.>>>

I disagree with this comment. ClanC and ClanR show statistically significant associations with the COI strain marker and with the host plant phenotype. See for example 250-274, Figure 6. It is true that the association is not absolute (there are exceptions) and this is discussed (376-387)

<<<In conclusion I think that the entire study would benefit with a reworking of context, should be rethought, conceptually and methodologically, based on the objective pursued. S. frugiperda is a pest of enormous economic importance, but to what extent both strains are associated with host plants is not clear. Studies from different approach are necessary to propose control and mitigation strategies for this species.>>>

I thank the reviewer for excellent comments and as can be seen there was a considerable reworking of the context of the paper that I believe addresses the stated concerns. 

Response to Reviewer 2:

>>>First, some of the conclusions drawn in this manuscript are not entirely supported by the data presented. For example, the title of this study comes from the conclusion on lines 337-338 that “this study confirms the dominant role of the Z-chromosome in strain divergence.” However, without looking broadly across the full genome to show that the Z-chromosome played a greater role in strain divergence than the autosomes, this statement cannot be supported. Given that data from only a single gene is presented here, this statement is an overreach that should be removed from the manuscript, and the title should be reconsidered.<<<

The title was changed as suggested. The argument for a prominent role of the Z-chromosome was also made more detailed though the conclusion was softened (Lines 359-376).

<<<Another concern is that the strain specific marker used through much of this study (Tpi exon) is closely linked to the region used to make the phylogenetic trees (Tpi intron). Therefore, the authors should be cautious about the conclusions drawn regarding strain relationships. For example, on lines 167-169, the authors state that the strain marker used on the Tpi exon is more reliable than the COI strain maker, because it more accurately matches their phylogeny. The support provided for this statement is good, however, this relationship could also simply be due to linkage. Based on the data, I don’t think this can be ruled out. The limitations of using these linked genomic regions should be explicitly stated in the discussion.>>>

The paper was rewritten and this concern about the linkage of the Tpi marker with the phylogeny taken into account. This linkage is now stated specifically as leading to the expectation of strong correspondence of the Tpi marker with the phylogenetic groups and serves as a standard for other comparisons (250-274). Please also see 81-94 for an explanation of the logic for using the intron segment.

<<<Finally, the current methods are not sufficient for publication. It is difficult to ascertain which genomic region was used in each analysis, and several population genomic analyses that are described in the results are not mentioned in the methods. The authors should add a section to the methods about their population genetic analyses, being sure to describe which programs were used and any necessary parameters. A few line-specific suggestions are listed below:>>>

These were addressed as described below for the specific comments.

<<<Title: Since only a single gene is assessed in this study, there is not enough data here to “confirm the fundamental role of the Z-chromosome in strain divergence.” The authors should change the title to highlight a better supported conclusion.>>>

Title was changed as suggested.

<<<Line 167-169: The phylogeny produced is not unbiased, but rather is linked to one of the markers being discussed. While the authors explanation is a valid alternative, linkage cannot be ruled out as driving this pattern.>>>

This section was rewritten as suggested (Lines 250-274). As noted above, this linkage is now stated specifically as leading to the expectation of strong correspondence of the Tpi marker with the phylogenetic groups and serves as a standard for other comparisons (250-274).

<<<Line 229: Remove the repeated word “up.”>>>

Done (line 198)

<<<Lines 255-279: These analyses are not described in the methods. Please add a section to the methodology to explain how these statistical tests were performed, which programs and parameters were used. >>>

This section was added in lines 467-486.

<<<Also, how were individuals assigned to groups in CS vs. CS and RS vs RS comparisons. Were these groups geographic or random?>>>

This is clarified by better labeling in the revised Figure (Fig 5C and Fig 5F).

<<<Line 273-274: How was it determined that the differences in signs for D and F* metrics were statistically significant? This should be added to the methods and justified.>>>

This was just poor writing and was corrected (lines 297-306). What was significant was the difference between the means of D and F* for the different comparisons.

<<<Line 290-291: The authors have not sufficiently shown that host choice is “the defining strain phenotype.” They can state this is an important strain phenotype, but the data seems to show that this phenotype is not strong enough to prevent inter-strain hybridization in C-strain habitats, especially in Florida.>>>

This phrasing was eliminated and replaced with an explanatory paragraph about the phenotype (lines 314-322). The importance of the host phenotype is demonstrated in Figure 5A-C and described in lines 205-237 and 336-345. 

<<<Line 303-311: While the authors can highlight these similarities, I recommend they also mention that their results are based on a single gene.>>>

This was done as seen in the paragraph 359-376.

<<<Line 337-349: As discussed above, this conclusion is not supported by the data presented. The authors should remove this paragraph from their discussion.>>>

The section was rewritten with a softer conclusion (“…support for the importance of Z-linked functions in determining the differential preference for plant hosts.” 416-417) and a more robust argument for the Z-chromosome (lines 359-376).

<<<Line 356-357: The data presented in this study does not support that host preference plays a “fundamental role” in strain divergence. Rather this study highlights a pattern, that can be ambiguous in some habitats and regions. Consider rewording this section.>>>

The section was extensively reworded 359-376.

<<<Lines 424-434: The authors only describe one method that was used in phylogeny creation.>>>

This was corrected. See Figure 4, 181-194. 

<<<The methods used to create figures 4 and 6 need to be detailed. >>>

A more detailed description is presented in lines 478-486.

<<<Also, it’s not entirely clear which segment of the gene was used for phylogeny creation. From the results, it seems like only TpiI4a200 was used, but there don’t seem to be primers listed specific to this region. Please clarify.>>>

Primers used are illustrated in Figure 2 and a more detailed and specific description for each segment is provided in lines 444-452

<<<Line 420-421: How many heterozygotes were removed and where were they found?>>>

Provided 501-505.

<<<Figure 3: Verify that all labels are correct in this figure. If I’m not mistaken, the CS and RS clade markers are reversed.>>>

Corrected and thank you for catching the error and the other excellent comments.

---

## [Decision Letter · Decision Letter 1]

26 Oct 2022

PONE-D-22-19114R1Observations of genetic differentiation between the fall armyworm host strainsPLOS ONE

Dear Dr. Nagoshi,

Thank you for submitting your manuscript to PLOS ONE. After careful consideration, we feel that it has merit but does not fully meet PLOS ONE’s publication criteria as it currently stands. Therefore, we invite you to submit a revised version of the manuscript that addresses the points raised during the review process.

We look forward to receiving your revised manuscript.

Kind regards,

Omaththage P. Perera, Ph.D.

Academic Editor

PLOS ONE

Journal Requirements:

Reviewers' comments:

Reviewer's Responses to Questions

**Comments to the Author**

1. If the authors have adequately addressed your comments raised in a previous round of review and you feel that this manuscript is now acceptable for publication, you may indicate that here to bypass the “Comments to the Author” section, enter your conflict of interest statement in the “Confidential to Editor” section, and submit your "Accept" recommendation.

Reviewer #2: All comments have been addressed

Reviewer #3: (No Response)

2. Is the manuscript technically sound, and do the data support the conclusions?

Reviewer #2: Yes

Reviewer #3: Yes

3. Has the statistical analysis been performed appropriately and rigorously? 

Reviewer #2: Yes

Reviewer #3: Yes

4. Have the authors made all data underlying the findings in their manuscript fully available?

Reviewer #2: Yes

Reviewer #3: Yes

5. Is the manuscript presented in an intelligible fashion and written in standard English?

Reviewer #2: Yes

Reviewer #3: Yes

6. Review Comments to the Author

Reviewer #2: The author has done an excellent job revising this manuscript and has addressed all of my original concerns. I only have a few minor comments that the author should consider before publication:

Line 270-273: I encourage the author to move this sentence down to around line 285. Currently, the authors talk about clan-host comparisons, then begin presenting findings about geographic differences, then go back to clan-host comparisons. This is not easy for the reader to follow. Consider revising this section to make these two ideas distinct.

Line 347: I agree that these two strains are experiencing differential selection pressures, but since this is correlative and not causative, it can not necessarily be linked to differential host use.

Supplementary Table S1: In the bottom of the table where C vs R comparisons are being made, the author should consider reordering the comparisons so that population 1 is all individuals of the C- strain and population 2 is the R-strain comparison. The data will be the same, but this would be a bit easier for the reader to interpret.

Reviewer #3: I thought this manuscript was extremely well written and clear with very few mistakes. The introduction set up FAW as an important pest around the world and an overview of the debate associated with host strain differentiation. I am not a population geneticist, but I was able to follow the rational for the analyses used and the interpretation of the results. Although only one gene was used as a marker, the author used multiple statistical approaches to determine its implications for splitting the FAW into two host strains. The figures did an excellent job at visualizing differences between host strains. The author also did a mostly thorough job at responding to previous criticisms.

General criticism

The abstract and introduction discussed the FAW as a new pest of the eastern hemisphere and that only one stain was found compared to the Americas. Does this study have any implications for to FAW in the eastern hemisphere? If not, it seems an unnecessary given the FAW is still a major pest in the Americas where this research was conducted.

How is this research different from other studies using the tpi gene? Lines 37-40 indicates other studies have used the tpi gene to differentiate between FAW strains.

How does this specific research further or alter the debate about the validity of calling these separate host stains? The study clearly advocates for the existence of different host strains, so is this study the end of the debate?

Specific criticism

Line 61: has should be have

Line 343: Reference needs to be numerical

Line 408: plant hosts should be host plants

Lines 418-420 say host use was only difference between host strains from Americas. Does this this contradict Lines 70-72 which mention differences in pheromones, reproduction, and mating between strains?

7. PLOS authors have the option to publish the peer review history of their article (what does this mean?). If published, this will include your full peer review and any attached files.

Reviewer #2: No

Reviewer #3: No

---

## [Author Response · Author response to Decision Letter 1]

27 Oct 2022

Reviewer #2

<<<Line 270-273: I encourage the author to move this sentence down to around line 285. Currently, the authors talk about clan-host comparisons, then begin presenting findings about geographic differences, then go back to clan-host comparisons. This is not easy for the reader to follow. Consider revising this section to make these two ideas distinct.>>>

The changes were made as suggested with lines 270-281 rearranged and modified. In addition an error was found in the numbering of the figures and this was corrected.

<<<Line 347: I agree that these two strains are experiencing differential selection pressures, but since this is correlative and not causative, it can not necessarily be linked to differential host use.>>>

Changed the necessary linkage to a speculative one by replacing “because” with “that could be related to” (line 345).

<<<Supplementary Table S1: In the bottom of the table where C vs R comparisons are being made, the author should consider reordering the comparisons so that population 1 is all individuals of the C- strain and population 2 is the R-strain comparison. The data will be the same, but this would be a bit easier for the reader to interpret.>>>

Done

Reviewer #3: 

General criticism

<<<The abstract and introduction discussed the FAW as a new pest of the eastern hemisphere and that only one stain was found compared to the Americas. Does this study have any implications for to FAW in the eastern hemisphere? If not, it seems an unnecessary given the FAW is still a major pest in the Americas where this research was conducted.>>>

Whether the strains exist and, if so, which are present in the Eastern Hemisphere has important ramifications to assessing crops at risk. Since this paper provides evidence supporting the legitimacy of the strain concept, I believe it is relevant to mention the situation in the Eastern Hemisphere. This rationale is noted in lines 33-35 and 37-39.

<<<How is this research different from other studies using the tpi gene? Lines 37-40 indicates other studies have used the tpi gene to differentiate between FAW strains.>>>

Past studies provided the empirical support for the use of Tpi SNPs as strain markers. The current study describes the phylogenetic relationships associated with the strains as defined by Tpi. This rationale is described in lines 81-92.

<<<How does this specific research further or alter the debate about the validity of calling these separate host stains? The study clearly advocates for the existence of different host strains, so is this study the end of the debate?>>>

This study provides genetic evidence supporting the existence of the strains and provides a phylogenetic description of how the strains differ. Whether it ends the debate on strains is for the scientific community to decide. 

Specific criticism

<<<Line 61: has should be have>>>

Done (line 62)

<<<Line 343: Reference needs to be numerical>>>

Done (line 320)

<<<Line 408: plant hosts should be host plants>>>

Done (line 412)

<<<Lines 418-420 say host use was only difference between host strains from Americas. Does this this contradict Lines 70-72 which mention differences in pheromones, reproduction, and mating between strains?>>>

No. Strain differences in pheromones, reproduction, and mating have been described, but only for specimens derived from North American populations. Lines 418-420 (now 389-390) states that “differential host use is the only trait demonstrated in host strain populations from both American continents”.

---

## [Editor Report · Decision Letter 2]

31 Oct 2022

Observations of genetic differentiation between the fall armyworm host strains

PONE-D-22-19114R2

Dear Dr. Nagoshi,

We’re pleased to inform you that your manuscript has been judged scientifically suitable for publication and will be formally accepted for publication once it meets all outstanding technical requirements.

Kind regards,

Omaththage P. Perera, Ph.D.

Academic Editor

PLOS ONE